# A Sensorless and Low-Gain Brushless DC Motor Controller Using a Simplified Dynamic Force Compensator for Robot Arm Application

**DOI:** 10.3390/s19143171

**Published:** 2019-07-18

**Authors:** Shih-Hsiang Yen, Pei-Chong Tang, Yuan-Chiu Lin, Chyi-Yeu Lin

**Affiliations:** 1Department of Mechanical Engineering, National Taiwan University of Science and Technology, Taipei 106, Taiwan; 2Ubiqelife Technology Corporation, Jhubei City, Hsinchu 302, Taiwan; 3Taiwan Building Technology Center, National Taiwan University of Science and Technology, Taipei 106, Taiwan; 4Center for Cyber-Physical System, National Taiwan University of Science and Technology, Taipei 106, Taiwan

**Keywords:** sensorless, Hall-effect sensors, dynamic compensator, low-gain control, low-pass filter, real-time control

## Abstract

Robot arms used for service applications require safe human–machine interactions; therefore, the control gain of such robot arms must be minimized to limit the force output during operation, which slows the response of the control system. To improve cost efficiency, low-resolution sensors can be used to reduce cost because the robot arms do not require high precision of position sensing. However, low-resolution sensors slow the response of closed-loop control systems, leading to low accuracy. Focusing on safety and cost reduction, this study proposed a low-gain, sensorless Brushless DC motor control architecture, which performed position and torque control using only Hall-effect sensors and a current sensor. Low-pass filters were added in servo controllers to solve the sensing problems of undersampling and noise. To improve the control system’s excessively slow response, we added a dynamic force compensator in the current controllers, simplified the system model, and conducted tuning experiments to expedite the calculation of dynamic force. These approaches achieved real-time current compensation, and accelerated control response and accuracy. Finally, a seven-axis robot arm was used in our experiments and analyses to verify the effectiveness of the simplified dynamic force compensators. Specifically, these experiments examined whether the sensorless drivers and compensators could achieve the required response and accuracy while reducing the control system’s cost.

## 1. Introduction

Industrial robots have been developed for decades and have served as a solution for the two automation insufficiencies: Quality uniformity and labor force. The number of robot arms in service applications is expected to exceed those in industrial applications by 2020 [1]. However, large-scale robot arm applications have not been observed in the service domain because their characteristics of quickness and high precision are accompanied by drawbacks of heaviness, low payload-to-weight ratio, and high risk. Robot arms used for service applications (hereafter referred to as service robots) do not require the same specifications as industrial robots; they instead emphasize mobility and safety of human–robot interaction [2]. In 2006, KUKA (Germany) launched the first lightweight robot arm, (LWR) [3]; the LWR’s elastic joint design [4], which incorporates torque sensors and provides safe control, has influenced various subsequent collaborative robot designs, such as the UR Robot (Universal Robots) and YuMi (ABB). However, despite the effective and safe control provided by the LWR design, its unique structure combining elastic joints and numerous sensors leads to high production cost, thus failing to achieve a favorable cost–performance ratio and limiting its service applications. A service robot does not require high-precision position control; thus, fewer sensors are required for service robots than for industrial robots. For example, replacing the encoders of position sensors with Hall-effect sensors can reduce the amount of additional sensor installation required; therefore, such robot arms exhibit lower production costs and wider potential applications in service domains.

Studies related to sensorless control of Brushless Direct Current (BLDC) motors have used Hall-effect sensors to perform motor position and speed control, although the low-resolution Hall sensors may result in signal undersampling. The majority of research, shown in Table 1, proposed the incorporation of low-pass filters in control as a solution for undersampling and noise but resulted in signal phase delay problems. The signal delay may be ignored or not apparent in the high-speed application, such as in [5]. In contrast, the delay issue in the low-speed application significantly reduced control accuracy. The compensator was an excellent way to solve the issue of signal delay. In Refs. [6,7,8,9], they designed compensator based on system models and experiments to achieve an excellent control effect in the simple or no-load system. However, this sensorless control study has not been applied in robot arm application, because their inherent problems of signal delay and low response prohibit the real-time control required by complicated dynamic robot systems.

Service robots share workspaces with their users; therefore, for safety at work, such robots should have limited velocity (<250 mm/s) and control force in robot motion [10]. Low-gain control is a safe control approach that avoids producing excessive unspecified forces, but low-gain control exhibits insufficient torque, which can lead to slow control response and low dynamic accuracy. Given a system model, the open-loop control approach can provide an excellent control response [11] and directly control motors in advance without any sensors providing status feedback. For example, current feedforward controllers can enhance the response of dynamic control. In Refs [12,13], they have incorporated robot dynamic equations into a robot controller to get better performance. However, these researches require a highly accurate system model, and the complex component configuration of robots yields great difficulty in obtaining the numerous system parameters, such as the moment of inertia and friction, etc. Moreover, the calculation of the dynamic equation involves complex multidimensional matrix operations, which requires long coding times and high costs, making it mostly infeasible on real-time robot control in reality.

Focusing on increasing control safety and reducing cost, we proposed a low-gain control architecture equipped with sensorless drivers and dynamic force compensators. In our previous research [14], we have presented a safety control method using dynamic compensators to detect an external force. In this study, the proposed architecture is intended to solve the problems of delayed signals in servo control and insufficient torque in low-gain control as well as to increase the dynamic control response and position accuracy. To address the complexity of system models, we employed a method to simplify dynamic equations and developed a controller to realize a seven-axis robot arm. Our experimental results proved the accuracy of the sensorless position control and confirmed that this control architecture achieves the real-time motion control effectiveness required of service robots.

## 2. Sensorless Control Problems

This study used sensorless drivers to control BLDC motors for low-cost robot application. Only Hall-effect sensors and a current sensor are used to perform position and current control. We have designed three joint modules with BLDC motors and harmonic drives according to the required torque (specifications presented in Table 2), with a minimum joint position resolution of smaller than 0.2°. To reduce the size of driver, this study used embedded drivers designed from a development board of Field-Programmable Gate Array (FPGA). The driver dimension is 80 × 65 × 35 mm^3^ with a double-layer power circuit board shown in Figure 1. Each driver controls two BLDC motors.

Hall sensors used in low-speed operation cause an undersampling problem, which results in discontinuous and high-frequency oscillation; thus, they are unsuitable for highly adaptive Proportional-Integral-Derivative (PID) controllers. Therefore, we propose in this chapter a low-pass filter capable of velocity adaptation to solve this problem.

### 2.1. Low-Speed Control with Low-Pass Filter

Sensorless current control was performed using a single-phase current sensing approach [15], which involved the use of only one current sensor and measuring phase Back Electromotive Force (EMF) to predict the current of the BLDC motors. However, this approach measured only the magnitude of current and could not detect current direction. To solve this problem, we adopted reference values based on output voltage directions, but this approach also exhibited a problem; when a current approached the zero-crossing point, the voltage led to high-frequency phase shifts, which resulted in incorrect current detection.

To obtain the motor velocity in sensorless control, the position feedback information must be changed for each sample. With an assumption that the motor’s control resolution is *n* [pulse/rev], the sampling frequency is $f_{s}$ [Hz], the minimum detectable velocity v [rpm] is given by (1). After substitution of detectable velocity range, which is 100–1000 rpm, and the Type 1 motor’s (Table 1) resolution into the equation, the sampling frequency must be 40–400 Hz.
(1)$$v=\frac{f_{s}}{n}\times 60$$

With the position resolution constant, this study changed the sampling frequency to solve these problems of speed and current detection [9]; we designed a Low-Pass Filter (LPF), which adjusted the sampling frequency $f_{s}$ according to the velocity command v, as presented in (2) where $k_{T}$ is sampling gain. The LPF could adapt to a wide range of velocity control and stabilize velocity signals by adjusting sampling frequency, also it can filter out high-frequency noises or oscillations from current signals.
(2)$$f_{s}=k_{T}\left|v\right|$$

This study incorporated three LPFs into the current controller and velocity controller to solve the signal discontinuity and high-frequency noise. The controller architecture on FPGA driver is shown in Figure 2. In current control, the LPF filtered out high-frequency noises and oscillations. Figure 3 compares the voltage signal outputs with and without the LPF. In velocity control, one LPF filtered the discontinuous signal on the velocity feedback and the other LPF smoothed the output signal on the velocity controller. Figure 4 compares the outputs with and without the LPF, thus producing more stable and comprehensive output signals in low-speed operation.

### 2.2. Phase Delay Problem

We added three LPFs in the closed-loop controllers to produce smooth and continuous output signals. However, these filters increase the sampling period and thus caused signal phase delay. Based on our experiment, the current output exhibited a phase delay of approximately 50–500 ms under different velocities. Such delay slowed the response of the closed-loop control and reduced the accuracy of dynamic control.

To accelerate the controller’s response, we added a velocity feedforward controller, which provided compensation through velocity commands, thus reducing the controller delay caused by the LPFs. Since the operation of the current feedforward controller required force information obtainable only from the robot arm motion, the current feedforward controller could not operate separately in each single-axis driver.

## 3. Simplified Robot Dynamic Compensator

Robot arms are nonlinear systems whose parameters are highly complex and difficult to obtain. Therefore, dynamic force calculation in a program is difficult and entails high costs and long calculation times. This chapter discusses the approaches we used to reduce development cost, namely simplifying the dynamic models and reducing the number of parameters, and how we increased the accuracy of the model parameters by experiments.

### 3.1. Robot Dynamic Model

The ideal dynamic model of an n-joint manipulator can be written in the Lagrangian form as [16]:(3)$$M\left(q\right)\ddot{q}+C\left(q,\dot{q}\right)\dot{q}+G\left(q\right)=\tau$$
where $q\in R^{n\times 1}$ is the joint variable vector [rad], $\tau \in R^{n\times 1}$ is the vector of generalized torque [N·m], $M\left(q\right)\in R^{n\times n}$ is the inertia matrix [kg$\cdot m^{2}$], $C\left(q,\dot{q}\right)\dot{q}\in R^{n\times 1}$ is the vector of Coriolis and centripetal torque [N·m], and $G\left(q\right)\in R^{n\times 1}$ is the vector of gravity torque [N·m]. Under single-joint motor steady-state operation [17], electromagnetic torque will be counter-balanced by load torque, inertia torque, and friction torque as Figure 5. Therefore,
(4)$$M_{m}{\ddot{\theta }}_{m}+B_{m}{\dot{\theta }}_{m}=\tau_{e}-\frac{\tau_{l}}{r}$$
(5)$$q=\theta_{s}=\frac{\theta_{m}}{r}$$
where $\tau_{e}$ is the electromagnetic torque, $\tau_{l}$ is the load torque [N·m], $M_{m}$ is the inertia constant of the rotor and coupled shaft [kg$\cdot m^{2}$], $B_{m}$ is the friction constant factor [N·m·s/rad], $\theta_{m}$ is the motor ration angle [rad], $\theta_{s}$ is the joint ration angle [rad], and r is the gear ratio.

Using (4) and (5), which expands calculation from a single-joint to n-joint, into (3), we obtain the motor dynamic Equation (6), where M(q), $C\left(q,\dot{q}\right)$, and G(q) are related to posture and motion of the robot arm, and $M_{m}\in R^{n\times n}$ and $B_{m}$
$\in R^{n\times n}$ are the constant matrices of motors. In the constant loading application, the dynamic inertia, Coriolis/centripetal, and gravity torque can be ignored. However, the serial robot arm has drastic changes in dynamic inertia and gravity torque with different postures. Therefore, the electromagnetic torque must be real-time calculated from each joint position feedback and robot arm model.
(6)$$\tau_{e}=\left(M_{m}+\frac{M\left(q\right)}{r^{2}}\right){\ddot{\theta }}_{m}+\left(B_{m}+\frac{C\left(q,\dot{q}\right)}{r^{2}}\right){\dot{\theta }}_{m}+\frac{G\left(q\right)}{r}\;$$

### 3.2. Simplified Robot Dynamic Model

The design of robot arms entails many parts of various materials and shapes, which reduce the accuracy of system model estimation [18], and requires numerous system parameters and a complex process to calculate dynamic force. To reduce the difficulty of obtaining and calculating system parameters, this study simplified a robot arm’s continuous mass model into mass points system according to its link system (Figure 6). To account for inertia and gravity torque calculation, this study adopted the Universal Robot Description Format (URDF) [19] to define the robot arms’ coordinate system (Table 3). To simplify relevant definitions in the URDF model, the coordinate directions of each joint are set to be the same directions as the world coordinate system. In Table 3, X, Y, and Z represent the offset of each joint coordinate, M refers to the mass of the link, CM denotes the offset of a link’s center of mass from the joint coordinate, and R gives the rotational directions of the motors. This study used a robot arm under a no-loading condition; the mass of Link 7 and center of mass could be adjusted if grippers or any other components were installed on the end-effector.

To calculate the moment of inertia, this study simplified the parallel axis theorem according to mass points system as shown in Figure 7, where $I_{P}$ is the inertia of point P, $I_{CM}$ is the default inertia based on the center of mass, such as that of a motor rotor and all structures, m is mass, and *d* is the offset between the link’s center of mass and point P. This moment of inertia was difficult to obtain through models; therefore, we conducted experiments to obtain these moments of inertia.

Dynamic inertia changes with the center of mass offsets; only the inertia component on the motor from offset ($d_{ki}$) is considered, which was projected from $\vec{r_{ki}}$ onto the motors’ rotation plane. Figure 8 shows an example for inertia calculation on Joint 1. Calculation of the two elements’ inertia was simplified as $k_{a0}+k_{a}M_{k}$, where $k_{a0}$ is a constant inertia coefficient, $k_{a}$ is a dynamic inertia coefficient, and $M_{k}$ is the dynamic inertia of $k^{th}$ joint, as presented in (7).
(7)$$M_{k}=\overset{7}{\underset{i=k}{\sum }}m_{i}\cdot d_{ki}{}^{2}=\overset{7}{\underset{i=k}{\sum }}m_{i}\cdot {\left|\vec{r_{ki}}\times \vec{\varepsilon_{k}}\right|}^{2}$$

Regarding the calculation of gravity torque, we calculate only the torque based on each link’s center of mass projection on the motors’ rotation direction. Specifically, gravity torque was simplified as $k_{g}G_{k}$, where $k_{g}$ is a gravity coefficient and $G_{k}$ is the gravity torque of *k*th joint. In Equations (7) and (8), $\vec{r_{ki}}$ refers to the vector from *k*th joint to *i*th link’s center of mass, $m_{i}$ is the mass of *i*th link, $\vec{g}$ denotes gravitational acceleration, and $\vec{\varepsilon_{k}}$ is the unit vector of *k*th joint’s rotating direction.
(8)$$G_{k}=\overset{7}{\underset{i=k}{\sum }}\left(\vec{r_{ki}}\times m_{i}\vec{g}\right)\cdot \vec{\varepsilon_{k}}$$

According to Equation (9), Coriolis and centripetal torques are related to the partial differentials of inertia [20]. In this study, the robot arm’s lightweight structural design produced relatively small inertia in its links, thus minimizing the influence on the arm’s overall motion. To favor calculation efficiency, this study disregarded the compensation for Coriolis and centripetal torques.
(9)$$C_{ij}\left(q,\dot{q}\right)=\frac{1}{2}\overset{n}{\underset{k=1}{\sum }}(\frac{\partial M_{ij}}{\partial \theta_{k}}+\frac{\partial M_{ik}}{\partial \theta_{j}}-\frac{\partial M_{kj}}{\partial \theta_{i}})\dot{\theta_{k}}$$

A friction identification experiment conducted by Wolf et al. [21] revealed that the friction in joint mechanisms was a complex nonlinear system; therefore, considering calculation efficiency, we employed a simplified friction Equation (10), where $\dot{q}$ is the motor angular velocity, $k_{v0}$ and $k_{v}$ are the coefficients of Coulomb friction and viscous friction, respectively. These two coefficients can be obtained through experiments.
(10)$$B_{m}=k_{v0}\mathrm{sgn}\left(\dot{q}\right)+k_{v}\dot{q}$$

Using the simplified process, we further simplified (6) as a new motor dynamic Equation (11), in which only dynamic inertia ($M_{k}$) and gravity ($G_{k}$) require kinematic model calculation; other simplified coefficients could be obtained through experiments as shown in Table 4. Accordingly, our simplification approach reduced calculation process and improved the efficiency of program execution.
(11)$$\tau_{d}=\left(k_{a0}+k_{a}M_{k}\right)\ddot{q}+k_{v0}\mathrm{sgn}\left(\dot{q}\right)+k_{v}\dot{q}+k_{g}G_{k}$$

### 3.3. Simplified Dynamic Force Compensator Design

In Equation (11), state information is represented by motor position (*q*), velocity ($\dot{q}$), and acceleration ($\ddot{q}$) and is usually obtained through the following two approaches.(a)Control commands: These calculate trajectories offline in advance and generate continuous and smooth signals [12,22]; however, they cannot reflect the real state of systems when excessively large errors occur.(b)Sensor feedback: This facilitates obtaining real-time state information but requires additional high-resolution sensors, such as accelerometers [23] and high-speed communication systems.This study employed only low-resolution position sensors, and thus approach (b) was not feasible. Moreover, in approach (a), the low-gain controller possessed large errors on differences between sensor feedback and command. To solve these problems, we proposed a hybrid approach (c).(c)Hybrid approach: A trajectory generator’s control commands are used to calculate velocity and acceleration. Position states are feedback through sensors to facilitate real-time position updates for each motor.

We substituted state information obtained through approach (c) into (11) and obtained (12), which yielded a Simplified Dynamic Force Compensator (SDFC), as presented in Figure 9 [13]. In this compensator, the calculations of dynamic inertia and gravity torque ($M_{k}$ and $G_{k}$) by forward kinematics, both of which require n-joint position information, are conducted separately in the robot controller with a period of 20 ms as the green block in Figure 9. Other compensation calculations are conducted in a FPGA driver operating with 1-ms interrupts. Under such a control system, the servo controller’s torque output, $\tau_{c}$, could be considered a, which is the sum of the external disturbance torque and the error of the compensator’s computed torques.
(12)$$\tau_{d}=\left(k_{a0}+k_{a}M_{k}\right){\ddot{q}}_{d}+k_{v0}\mathrm{sign}\left({\dot{q}}_{d}\right)+k_{v}{\dot{q}}_{d}+k_{g}G_{k}$$

## 4. Simplified Dynamic Force Compensator Tuning

In the SDFC design, we simplified system models by reducing the number of system parameters; specifically, the robot controller calculates only the dynamic inertia and gravity torque by robot posture model, and the FPGA driver facilitates adjustments in compensation gain through experiments, which generate results more applicable to actual systems. The following sections discuss the experimental design used to adjust the gain parameters in the SDFC.

### 4.1. Simplifying the Calculations of Moment of Inertia and Gravity Torque

Equations (7) and (8) expedited our calculations of dynamic inertia and gravity torque ($M_{k}$ and $G_{k}$), but the resulting floating-point number was excessively large for real-time communication and could not be efficiently processed in the integral calculation of DSP in the FPGA driver. Therefore, we normalized the calculated inertia and gravity, namely multiplying the calculated results by proportional coefficients ($P_{M}$ and $P_{G}$) to transform these results into integers from −100 to 100. Taking the seven-axis robot arm as an example, the maximum result of each joint and proportional coefficients are presented in Table 5. With no-load on end-effector, there was no dynamic inertia and gravity torque on Joint 7. Proportional errors resulting from normalization could be reduced by adjusting the gains ($k_{a}$ and $k_{g}$) through experiments. Finally, the dynamic inertia and gravity torque are included in real-time control commands, such as position and velocity command, between the robot controller and FPGA drivers for updating real-time robot status.

### 4.2. Tuning Experiment with SDFC

To adjust the compensator’s gain, this study designed three experiments to measure the currents when dynamic forces occur. We analyzed gravity, friction, and inertia torques separately for each joint and adjusted the gain to generate a compensatory force that approximates the actual force of real robot systems. The experimental steps for each joint were as follows:(a)Gravity compensator tuning: the force resulting from gravity acting upon the robot arm changes only with its posture; therefore, we first adjusted the gravity compensator. In the experiment, we adjusted the robot joints to a posture that yielded the greatest gravity torque and obtained a computed gravity parameter $G_{k}$ of 100. A stable motor current $i_{g}$ under this posture was measured and substituted into (13), which yielded a gravity gain $k_{g}$.
(13)$$i_{g}=k_{g}G_{k}$$(b)Friction compensator tuning: after adjusting the gravity compensator, a gravity torque can be disregarded in control processes; thus, the next step is friction compensator adjustment. Because friction only occurs during the motion of motors and is positively related to velocity, we conducted controlled experiments with constant velocity to measure servo currents and obtained average currents $i_{f}$ under different velocities as the blue dots in Figure 10. Subsequently, the results were used to conduct a linear regression analysis, which yielded two linear equations, namely the red lines in Figure 10. Using (14), the two friction coefficients $k_{v}$ and $k_{v0}$ could be obtained from the regression line as the slope and the y-intercept, respectively. Finally, the gain coefficients of the two directions were averaged and incorporated into this step, and this process was repeated for further adjustment.
(14)$$i_{f}=\left[k_{v0}sgn\left(\dot{q}\right)+k_{v}\dot{q}\right]$$(c)Inertia compensator tuning: inertia force occurs only during the acceleration and deceleration stages; because these two stages usually have short durations (<1 s) and suffer from current signal delay, accurate current values are rarely obtained. Therefore, this study used a numerical approximation to adjust inertia gains ($k_{a}$ and $k_{a0}$) rather than using currents to predict them. First, we generated a trajectory for a given acceleration. Under low-inertia conditions (mainly default inertia), $k_{a0}$ was adjusted to reduce the errors of servo currents in the acceleration and deceleration intervals; under high-inertia conditions (mainly dynamic inertia), $k_{a}$ was adjusted to the same step as $k_{a0}$ adjustment. These two steps facilitated the inertia calculation in (15) to more accurately approximate real systems.
(15)$$i_{\mathrm{inertia}}=\left[k_{a0}+k_{a}M_{k}\right]\ddot{q}$$

Comprehensive force compensation was achieved after the three aforementioned adjustment steps. For example, the position control results in Joint 2 is illustrated in Figure 11, in which the red line represents SDFC’s output current, the blue line represents the servo controller’s output current, and the black line represents the sum of these two’s output currents. Because the SDFC provided the needed currents for motion, the servo controller was required only to reduce position errors; this greatly reduced servo controller current output. The tuning experiments yielded the compensation gains for the seven-axis SDFC; these are listed in Table 6.

### 4.3. Analysis of Phase Compensation

Before the incorporation of the SDFC, the phase delay in the control signals caused by the low-gain controller and LPFs resulted in low-accuracy position control. For example, Figure 12 presents the result of Joint 1’s position control (velocity of 60°/s) and reveals that the closed-loop servo control without SDFC leads to a delay of 50 ms and position error greater than 10° (blue line). After incorporating the SDFC, the force compensator reduced the dynamic error to less than 1° (red line). Comparing the current output results with and without the SDFC, current response was faster with compensation; this verified that the SDFC increased control response and reduced dynamic error.

## 5. Real-Time System Framework

To achieve an efficient control effect and dynamic compensations, a robot controller must transmit control information to each motor driver in real-time. The control frequency is usually set higher than 1 kHz, and the controllers must be equipped with a Real-Time Operating System (RTOS) and real-time communication to avoid system delays, which could cause discontinuity and further delays.

This study used open-source software to design a real-time robot controller prioritizing low cost and employed the Raspberry Pi 3 Module B (Raspberry Pi Foundation, Cambridge, UK) and Linux Ubuntu MATE system accompanied by a Linux Xenomai 3 RTOS to establish a real-time platform. This chapter explains the real-time architecture of our controller.

### 5.1. RTOS–Xenomai

Xenomai [24] is an open-source RTOS provided on the Linux platform, uses a dual kernel structure and Adaptive Domain Environment for Operating Systems (ADEOS) to manage cokernel and user-kernel boundaries, which have high priority, and schedules real-time and non-real-time tasks in CPU processing. While other RTOSs [25] focused on the lowest technically feasible latencies, Xenomai also considers clean extensibility (RTOS skins), portability, and maintainability as very important goals.

Compared with a previous version, Xenomai 2.6, Xenomai 3 has improved performance of the Cobalt kernel, which reduces latency to 10 μs on a CPU-stressed setup. It provides more skins and classifies different application programming interface libraries according to their applications; it enables users to select different skins according to their needs and simplify and expedite program design. Regarding this study’s real-time controller, we used the Xenomai’s Native skin to design a real-time motion control program. To facilitate real-time communication, we used the Real-time Drive Model (RTDM) skin [26] to design a real-time communication architecture.

### 5.2. Real-Time Communication

A standard CANopen system (an industrial control network) [27] is equipped with two real-time communications, namely CANbus and EtherCAT. In particular, CANbus exhibits low cost and high anti-interference but has a short data length of 16 bytes; therefore, it is not suitable for real-time multi-node control. EtherCAT demonstrates the advantage of high information throughput, indicating its suitability for real-time multi-node control, but is costly in its hardware and communication chips.

Considering the aforementioned factors and attempting to build controllers and drivers into the robot arm to reduce communication distance, we applied a Serial Peripheral Interface (SPI) bus between a robot controller and FPGA drivers, designed a communication system, and simplified transmitted data packets that contained synchronous and asynchronous transmissions such as each’s position, velocity, and current.

Regarding SPI driver design, Linux provides SPI bus libraries; however, as a non-real-time operating system, Linux resulted in an SPI communication frequency at 1 kHz, a jitter longer than 10,000 μs, and an average latency of 10 μs (Figure 13 [top]). The Linux drivers exhibited excessively long latency and thus were not appropriate for robot systems requiring 1-ms control response. To achieve real-time control, this study used Xenomai’s RTDM skin to design a real-time SPI driver, which had a jitter shorter than 100 μs and an average latency of 0.5 μs (Figure 13 [bottom]). These features indicated the SPI driver was appropriate for application in the communication architecture of a real-time controller.

### 5.3. Real-Time Program Architecture

Focusing on controllers’ processing efficiency, we have separated their program tasks according to their level of priority of real-time control. The non-real-time Linux kernel was responsible for low priority tasks such as kinematic calculations, trajectory generation, and user interface display. The Xenomai kernel was responsible for real-time SPI bus communications, control command transmission, state information feedback, and synchronization of each joint’s motor state with the drivers with a control period of 1 ms.

A bridge program between the two kernels was designed to facilitate inter-process communication; namely, programs in the two kernels could operate independently during different periods, and their program design could be developed independently. An FPGA-based DSP control system was used to conduct 1-ms servo control and dynamic force compensation. In the first joint, SPI communication was used between its driver and the robot controller, and RS485 serial communication was used between its driver and other joints’ drivers. The overall communication and control architecture is illustrated in Figure 14.

## 6. Experiment and Results

This study used a self-developed seven-axis robot arm, which exhibited concise design with low-functionality components, to conduct experiments to verify the validity of this study. The specifications and prices of the main components used in the robot arm’s joints are listed in Table 7.

### 6.1. Joint Position Accuracy Analysis

Different motors and reduction ratios used in the joints resulted in their various minimum resolutions (Table 2), with each of the joints achieving a position resolution of less than 0.2°. To verify the control effectiveness of the SDFC, we conducted experiments to test both single-axis and multiple-axis position accuracy.

#### 6.1.1. Experiment 1: Single-Axis Repeatability Accuracy

Within each joint’s working range (Table 8), we conducted position control by randomly generating 20 positions under an acceleration duration of 0.5 s, deceleration duration of 1 s, and speed range of 20–100% (an interval of 10% and maximum angular velocity of 60°/s). Figure 15 depicts the position errors of each joint, and Table 8 presents the mean absolute errors (MAE) and standard deviations (SD) of each joint.

#### 6.1.2. Experiment 2: Multiple-Axis Repeatable Accuracy

The following four positions with different loads were selected:
P1 = [0°, −30°, 0°, −60°, 0°, −90°, 0°]P2 = [30°, −60°, 30°, −60°, 30°, −60°, 60°]P3 = [0°, −90°, 0°, 0°, 0°, 0°, 0°]P4 = [60°, −60°, 30°, −90°, −45°, −40°, −60°]

Figure 16 shows these four positions’ resulting robot arm postures (P1 to P4). This study used linear interpolation to generate each joint’s trajectories, according to which we posed the robot arm in the four positions under a maximum velocity of 60°/s and recorded the steady-state position errors. The process of position control and recording of steady-state position errors has been repeated for 100 cycles. Figure 17 demonstrates mean absolute errors of each position; Table 8 shows the mean absolute errors and standard deviations of each joint.

According to Experiment 1, Joint 2 exhibited the lowest accuracy and had an error of 0.23°, which was larger than the minimum resolution (0.1875°). This is because, of all joints, Joint 2 had the highest load, which impeded its fine movements and position convergence, thus failing to effectively eliminate steady-state errors. All joints other than Joint 2 had errors of approximately 0.1°, which was smaller than the minimum resolution under different velocity controls.

In Experiment 2, because the inertia and gravity in each joint changed concurrently, mutual force between them occurred, and thus the position errors were larger than those of single-axis control. In particular, Joints 2 and 4 had the largest steady-state errors; this result is because the two joints carried the largest loads of all. According to the standard deviations in Table 8, all joint position errors were within 0.5° (about 95% of data fall within two standard deviations of the mean), equal to 1~2 times the minimum resolution. The results of two experiments verified that the SDFC helped the low-gain controllers reduce dynamic error under different motor speeds and loads.

### 6.2. Compensator Efficiency Analysis

According to the study’s control structure, the motor actual current output was the sum of servo controllers’ and the SDFC’s torque, as shown in Figure 6. High similarity between a compensator’s calculated dynamic force and the actual force received indicates a reduced servo control output current produced by errors. Therefore, the servo control current can be seen in the SDFC compensating for error; a low proportion of servo current in the total output current (16) indicated high accuracy of the SDFC, where *E* is current error rate, $I_{\mathrm{servo}}$ is the servo controller current, and $I_{\mathrm{out}}$ is the total motor output current.
(16)$$E=\left|\frac{I_{\mathrm{servo}}}{I_{\mathrm{out}}}\right|\times 100\%$$

#### Experiment 3: Compensator Efficiency Analysis

This experiment separated the motion process into acceleration, period I; constant velocity, period II; and deceleration, period III (Figure 18). The acceleration period involved friction, gravity, and inertia torque; the constant velocity involved only friction and gravity torque. The current error rate (*E*) in the three periods were used to analyze the accuracy of compensators’ force calculation relative to actual force received.

The test condition for each joint comprised a velocity of 60°/s and a position control range of 0° to 90°. During the experiments, we recorded each joint’s total output current, the SDFC’s currents, and the servo controllers’ currents; subsequently, we calculated the current error rate (*E*) from (16) and average currents in the three motion periods to evaluate the efficiency of the compensator. Table 9 presents the experimental results.

According to the results of Experiment 3, Joint 2 had a relatively large servo current output in the constant velocity period and exhibited a current error (*E*) of approximately 40%. This is because the controlled current crossed the zero point during period II (Figure 18), which produced large oscillations in current control. Accordingly, improving current control in the zero-crossing point can possibly reduce such control errors.

Among the three motion periods, Period I demonstrated the highest current error rate (>10%) because the static friction must be overcome to activate the motor, and the friction compensator as (10) cannot accurately calculate the static friction for motor starting; therefore, a large control error occurred. The remaining periods all exhibited a current error rate of less than 10%, indicating that the accuracy of SDFC in force calculation relative to the actual forces received was higher than 90%; therefore, the SDFC had adequate resolution for robot arm dynamic control.

According to the result in Table 9, all joints’ average servo currents were smaller than 0.2 A, which is less than 10% of the total current output. The result verified that the force compensation provided by the SDFC in dynamic control could correct more than 90% of all errors, whereas other errors, such as calculation errors and system model errors, could be corrected using closed-loop servo controllers. The integrated use of a dynamic force compensator and servo controllers provided favorable control effectiveness without the use of a high-precision system model.

## 7. Conclusions

This study used sensorless drivers and SDFCs to design and control a seven-axis robot arm. Through open-loop current compensation, we solved the servo controllers’ delay problem caused by an insufficient number of sensors and the problems of insufficient force and slow response caused by low-gain control. The SDFC was designed by simplifying calculations and conducting experiments to obtain the needed parameters without the high-precision system model, which overcame problems occurring when using only Hall-effect sensors, and simplified the dynamic control calculation steps to achieve more programmable calculation with the use of low-cost processors. Moreover, because the compensator is separate from the servo controllers, it can be applied to various servo systems and robot arms with only some modifications in the system model parameters, thus exhibiting high flexibility of use and wide applicability. The compensator could also reduce the system loads of servo control as well as the performance requirements of a motor driver and sensor, thus reducing the development cost of robot arms. The results presented 0.2° accuracy of joint position control and 90% real-time current compensation from SDFCs; these performances were good enough for service application. In addition, motors with many poles, such as Maxon Motor’s flat motors with 11 poles, can be used to increase position resolution from the 0.188° achieved by this study to approximately 0.068°. Overall, this study verified the safe and accurate control of the proposed sensorless, low-gain driver and thus its applicability in service robots.

## Figures and Tables

**Figure 1 sensors-19-03171-f001:**
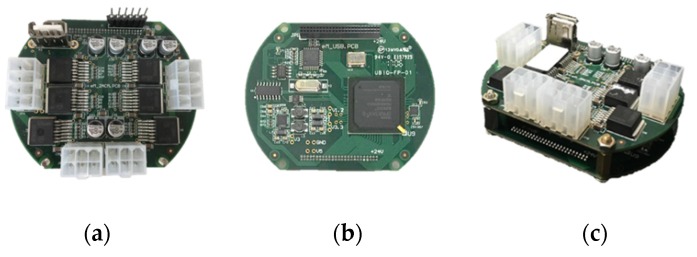
Embedded Field-Programmable Gate Array (FPGA)/Digital Signal Processor (DSP) driver: (**a**) layer1; (**b**) layer2; (**c**) assembled driver.

**Figure 2 sensors-19-03171-f002:**
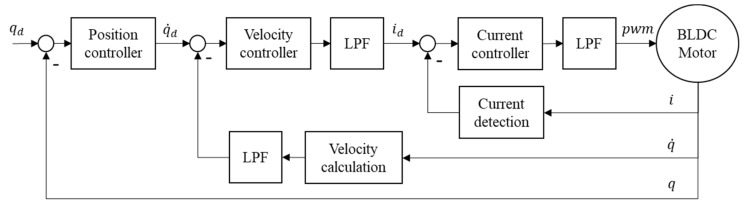
Servo control diagram on FPGA driver.

**Figure 3 sensors-19-03171-f003:**
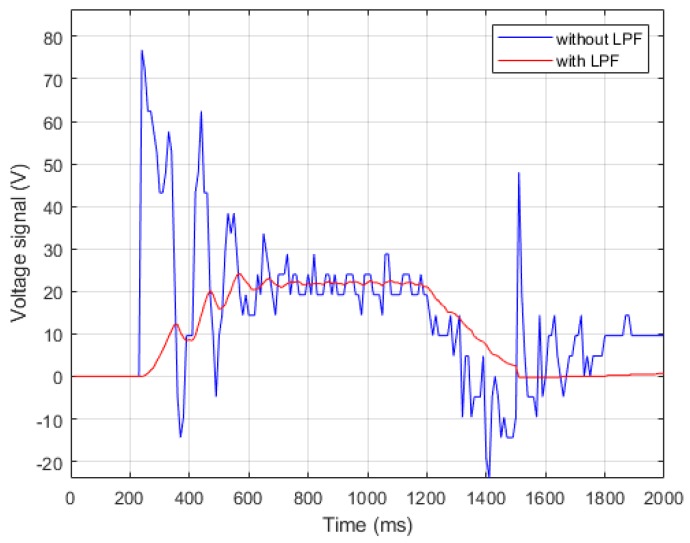
Voltage signal output on current controller.

**Figure 4 sensors-19-03171-f004:**
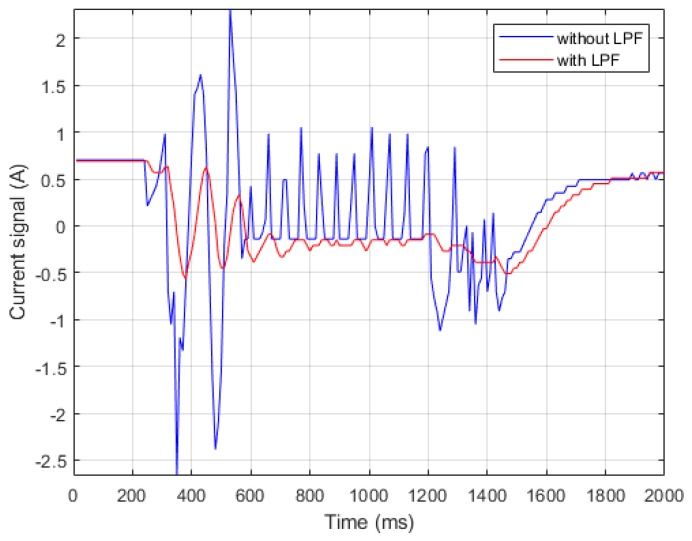
Current signal output on velocity controller.

**Figure 5 sensors-19-03171-f005:**
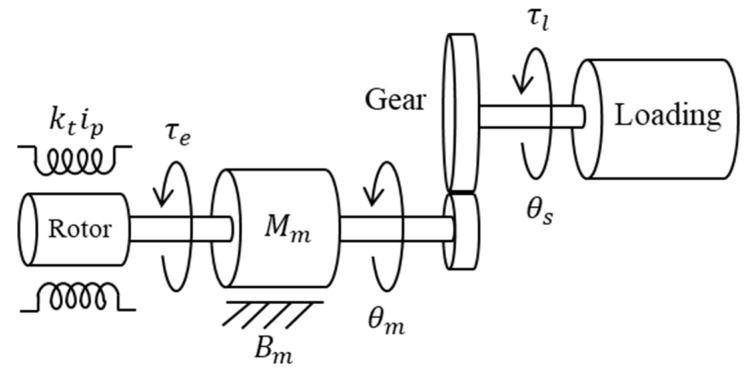
Lumped model of a single link with actuator/gear train.

**Figure 6 sensors-19-03171-f006:**
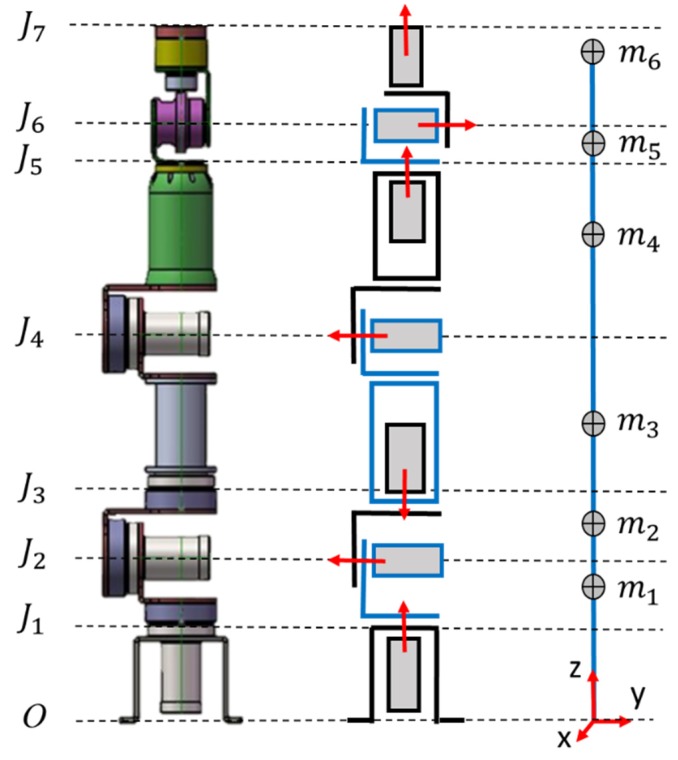
Simplified mass-point model on a seven-axis robot.

**Figure 7 sensors-19-03171-f007:**
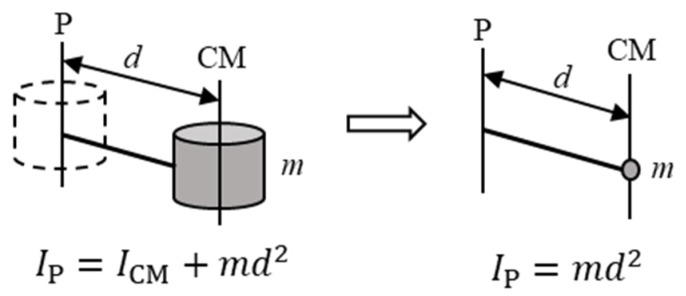
Simplification of the parallel axis theorem.

**Figure 8 sensors-19-03171-f008:**
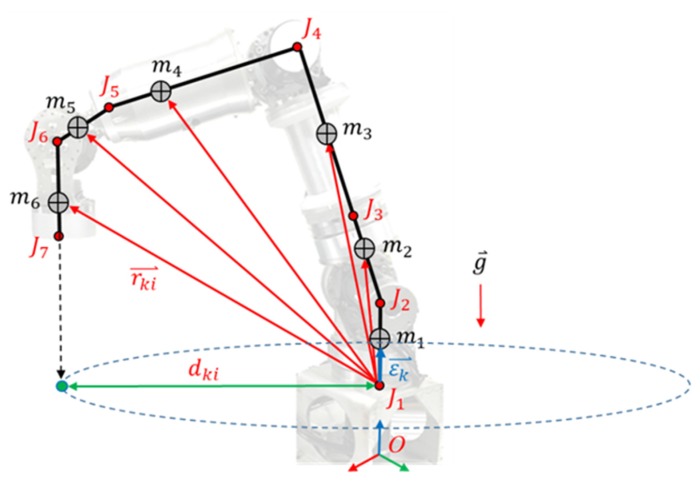
Dynamic inertia calculation of *k*th joint (example for *k* = 1, *i* = 6).

**Figure 9 sensors-19-03171-f009:**
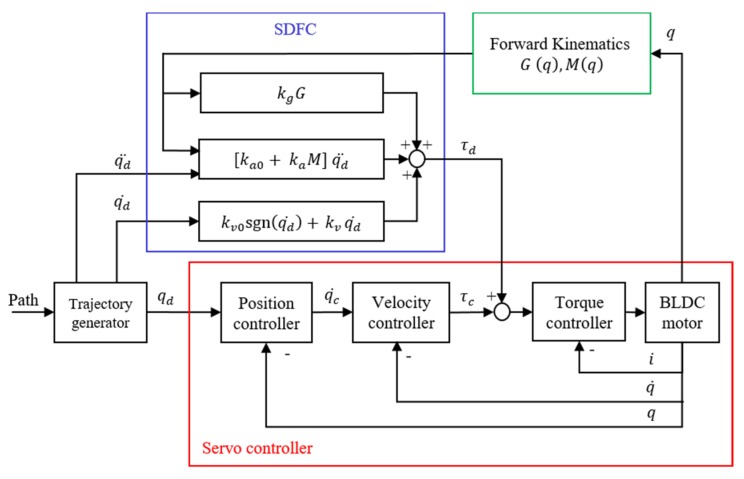
Simplified Dynamic Force Compensator (SDFC) control diagram.

**Figure 10 sensors-19-03171-f010:**
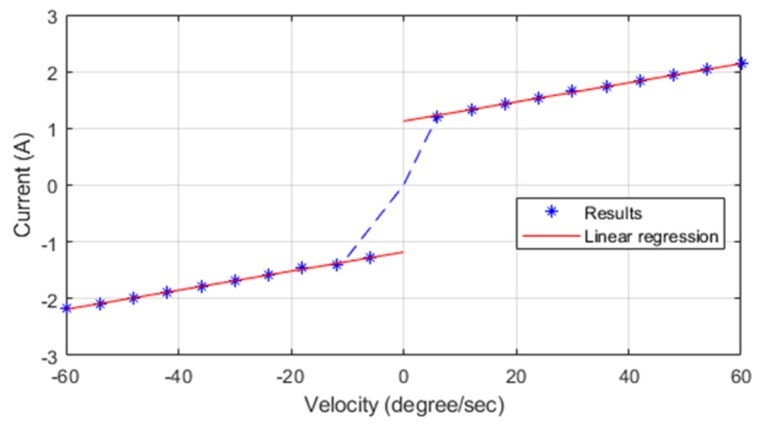
Friction identification experiment.

**Figure 11 sensors-19-03171-f011:**
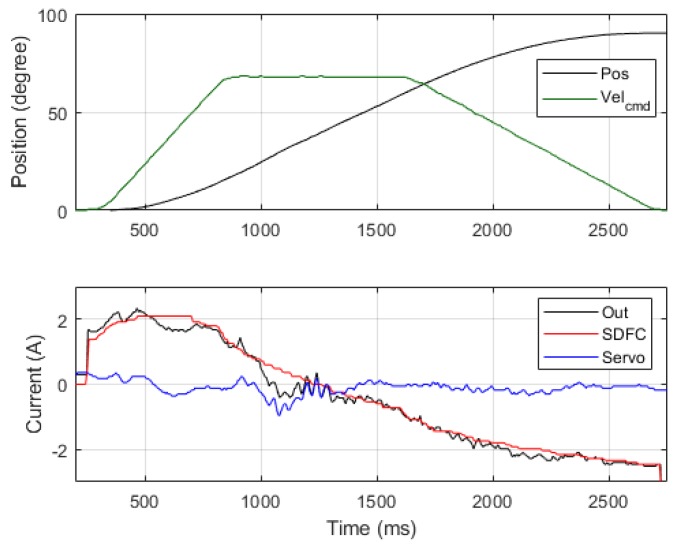
Current output performance for Joint 2 with an SDFC.

**Figure 12 sensors-19-03171-f012:**
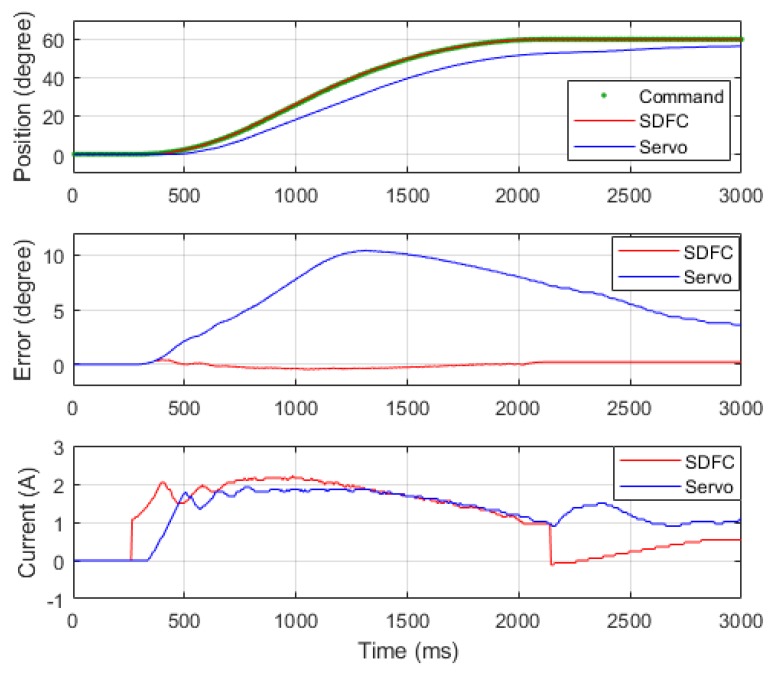
Results comparison with SDFC and closed-loop servo controller.

**Figure 13 sensors-19-03171-f013:**
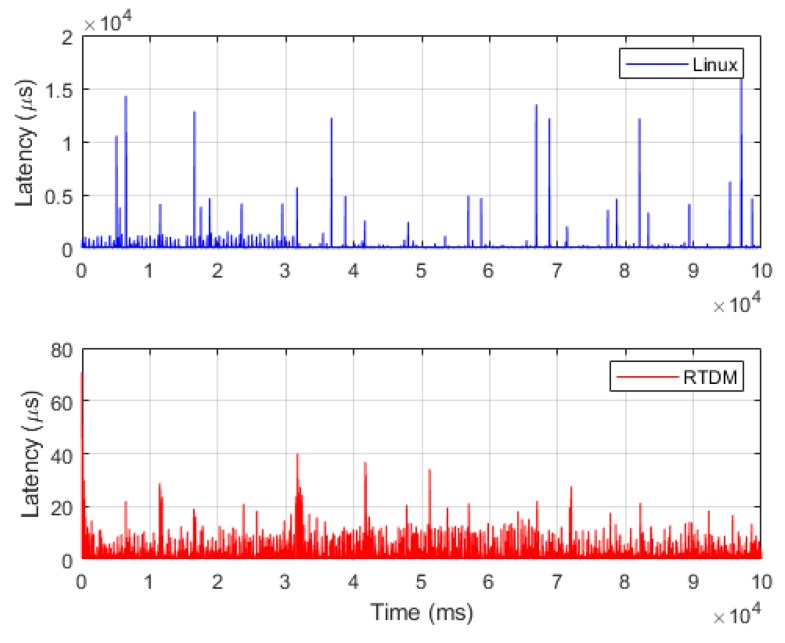
Latency performances of Serial Peripheral Interface (SPI) driver.

**Figure 14 sensors-19-03171-f014:**
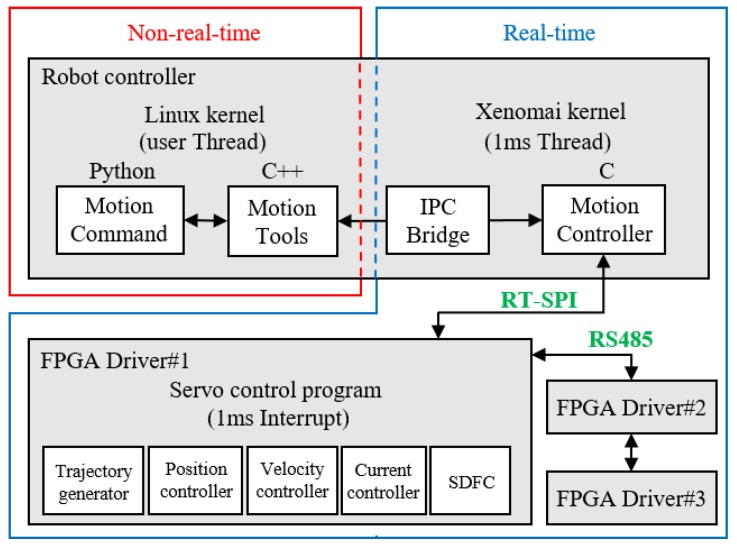
Real-time control architecture.

**Figure 15 sensors-19-03171-f015:**
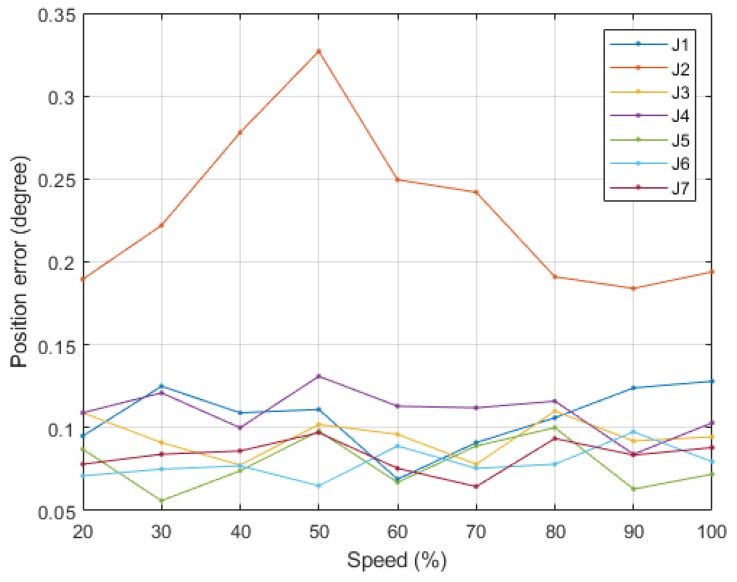
Position error results of experiment 1.

**Figure 16 sensors-19-03171-f016:**
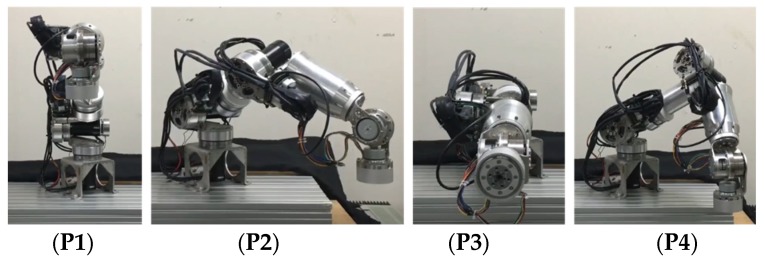
Robot arm postures using multiple-axis control.

**Figure 17 sensors-19-03171-f017:**
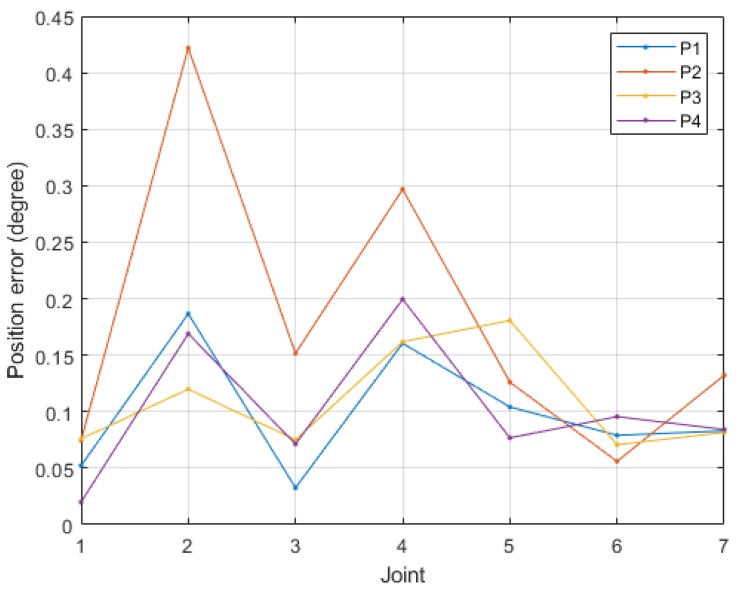
Position error results of experiment 2.

**Figure 18 sensors-19-03171-f018:**
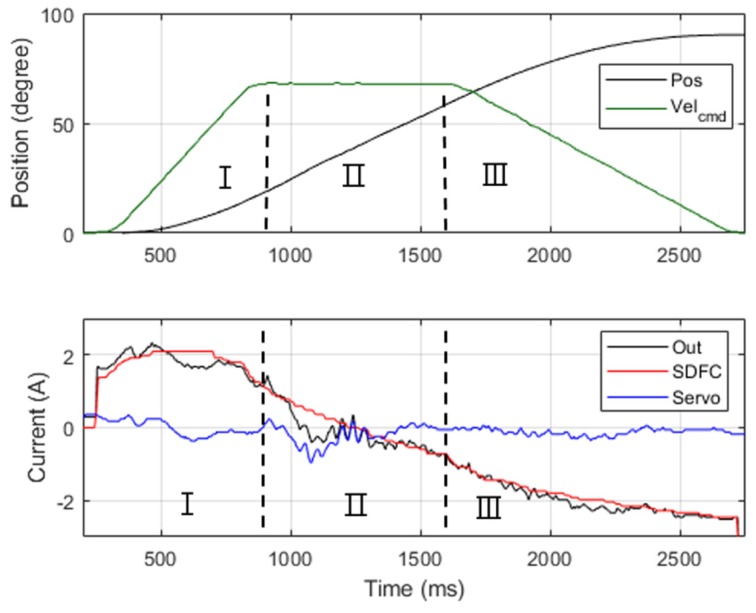
Dynamic current performance for Joint 2.

**Table 1 sensors-19-03171-t001:** Comparison of sensorless Brushless Direct Current (BLDC) motor control research.

References	[6]	[7]	[8]	[9]	Proposed Approach
Speed range (rpm)	19–4600	20–1000	1000–4800	200–3000	100–1000
Filter	Low-pass filter	Low-pass filter	Low-pass filter	Low-pass filter	Low-pass filter
Motor poles	4	4	4	4	4
Application	No load	Petroleum drilling system	Driveline system	No load	Robot arm
Solution of phase-delay	Flux linkage threshold	Adaptive compensation	Active compensation	Novel speed calculation	Dynamic compensation

**Table 2 sensors-19-03171-t002:** Specification of 3-phase BLDC motors.

Parameter	Type 1	Type 2	Type 3
Joint	J1–J4	J5/J7	J6
Weight	1050 g	150 g	150 g
Voltage	24 V	24 V	24 V
Rated speed	1000 rpm	4840 rpm	4840 rpm
Rated current	3.6 A	3.26 A	3.26 A
Rated torque	0.6 N·m	0.13 N·m	0.13 N·m
Torque constant	0.16 N·m/A	0.037 N·m/A	0.037 N·m/A
Motor poles	4 pairs	8 pairs	8 pairs
Resolution	24 pulse/rev	48 pulse/rev	48 pulse/rev
Gear rate	1:80	1:50	1:120
Min. joint resolution	0.1875°	0.15°	0.0625°
Max. joint speed	75°/s	576°/s	242°/s

**Table 3 sensors-19-03171-t003:** Universal Robot Description Format (URDF) table of a seven-axis robot arm.

Joint	X [mm]	Y [mm]	Z [mm]	M [kg]	CM [mm]	R [axis]
J1	0.0	0.0	156.0	2.95	31.0	Z
J2	0.0	0.0	58.5	2.17	43.7	−Y
J3	0.0	0.0	63.5	4.35	147.7	−Z
J4	0.0	0.0	229.5	2.43	86.6	−Y
J5	0.0	0.0	222.4	1.24	48.5	Z
J6	15.0	0.0	53.0	1.07	70.5	Y
J7	0.0	0.0	110.7	0.0	0.0	Z

CM = Center of Mass.

**Table 4 sensors-19-03171-t004:** Coefficient definition on the dynamic equation.

Physical	Coefficient
Default moment of inertia	$k_{a}$
Dynamic moment of inertia	$k_{a0}$
Viscous friction	$k_{v}$
Coulomb friction	$k_{v0}$
Gravity torque	$k_{g}$

**Table 5 sensors-19-03171-t005:** Maximum dynamic inertia and gravity torque.

Joint	$M_{k}$ $\left[\mathrm{kg}\cdot m^{2}\right]$	$P_{M}$	$G_{k}\;[\mathrm{kg}\cdot m]$	$P_{G}$
J1	1.3790	72.52	0	0
J2	1.3795	72.50	3.3165	30.15
J3	0.2377	420.70	0.9175	108.99
J4	0.2382	419.64	0.9175	108.99
J5	0.0081	12,345.68	0.1090	917.43
J6	0.0054	18,518.52	0.0758	1,319.26

**Table 6 sensors-19-03171-t006:** SFDC compensation gain.

Joint	$k_{g}$	$k_{v}$	$k_{v0}$	$k_{a}$	$k_{a0}$
J1	0	40	145	200	50
J2	−1250	40	180	150	100
J3	−230	50	220	150	67
J4	−230	50	200	100	100
J5	−120	10	100	10	20
J6	−50	35	80	40	50
J7	0	30	160	0	30

**Table 7 sensors-19-03171-t007:** Prices and specifications of the main joint components.

Item	Type	Source	Unit Price
Motor	3-phase BLDC motor	China	35 USD
Reduction gear	Harmonic driver	Taiwan	400 USD
Driver	FPGA build-in DSP	Taiwan	100 USD
Total price for 7 axis robot arm		5000 USD	

**Table 8 sensors-19-03171-t008:** Results of experiment 1 and experiment 2.

	Work Range		Experiment 1		Experiment 2	
Joint	Min.	Max.	MAE	SD	MAE	SD
J1	−90°	90°	0.11°	0.12°	0.06°	0.06°
J2	−30°	90°	0.23°	0.33°	0.23°	0.24°
J3	−90°	90°	0.09°	0.10°	0.08°	0.07°
J4	−90°	90°	0.11°	0.10°	0.20°	0.07°
J5	−90°	−90°	0.08°	0.09°	0.12°	0.20°
J6	−90°	90°	0.08°	0.09°	0.08°	0.06°
J7	−360°	360°	0.08°	0.10°	0.09°	0.17°

MAE = Mean Absolute Error. SD = Standard Deviation.

**Table 9 sensors-19-03171-t009:** Results of experiment 3.

	Current Error Rate *E* [%]			Current [A]	
Joint	I	II	III	Servo	Total
J1	8.56	2.01	2.18	0.07	1.67
J2	10.67	39.68	6.08	0.17	1.51
J3	10.64	8.12	4.48	0.16	2.32
J4	12.12	4.49	4.99	0.13	1.88
J5	7.26	1.51	6.49	0.05	0.91
J6	11.80	4.34	3.71	0.06	1.07
J7	8.22	4.00	3.99	0.08	1.41

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
