# Peer review of "A Sensorless and Low-Gain Brushless DC Motor Controller Using a Simplified Dynamic Force Compensator for Robot Arm Application"

_sensors, 2019, doi:10.3390/s19143171_

Round 1

Reviewer 1 Report

This paper is similar to another paper by the same authors, but expands on previous ideas with the addition of multiple real world applications and tests.

I am unsure if the previous paper has been published yet, but it should be cited as a part of this paper, as I believe it uses the same or a similar arm to the one developed in the earlier paper.  In addition, Figure 7 in this paper is identical to an image in the earlier paper.

This paper adds new dynamic models, and a variety of interesting experiments.

There are some minor typos/english writing issues to be addressed.  For example, Section 3 is called 'Simplify Robot Dynamic Compensator', and should read 'Simplified Robot Dynamic Compensator'.

This paper does a nice job explaining experiments and results, significantly improved from the last submission I viewed by these authors.

One weakness I see is that the results should more clearly be tied to conclusions.

Please make a careful check of english writing and grammar, and add necessary citations as noted.

Author Response

Dear reviewers:

We are very grateful to the precious comments made by reviewers, which provide insightful directions for significantly improving the quality of the manuscript. According with these advices, we carefully amended relevant parts in the manuscript. The responses to the comments and questions are listed below.

To 1st reviewer

Point 1:

I am unsure if the previous paper has been published yet, but it should be cited as a part of this paper, as I believe it uses the same or a similar arm to the one developed in the earlier paper.  In addition, Figure 7 in this paper is identical to an image in the earlier paper.

Response 1:

Thank you for your suggestion. Our previous paper has been published on June 5. The previous research work has been used to improve results and the implementation details on motor control are discussed. Following your suggestion, the reference of the previous paper have been cited. (P. 3, Line 78-80 and P.10, Line 235).

Point 2:

This paper adds new dynamic models, and a variety of interesting experiments. There are some minor typos/English writing issues to be addressed. For example, Section 3 is called 'Simplify Robot Dynamic Compensator', and should read 'Simplified Robot Dynamic Compensator'.

Response 2:

Thank you for your correction. It was our mistake of writing on the title of Section 3 and Section 4; we have revised to the correct title “Simplified Robot Dynamic Compensator” based on your suggestion. (P. 6, Line 145 and P. 10, Line 244)

Point 3:

This paper does a nice job explaining experiments and results, significantly improved from the last submission I viewed by these authors. One weakness I see is that the results should more clearly be tied to conclusions.

Response 3:

Thank you for your suggestion. We think our results are better than past researches on sensorless robot control. The experiment result presented 0.2 accuracy of joint position control and 90% real-time current compensation from SDFCs; these performances are good enough for service applications. We have improved this discussion of results in conclusions. (P. 20, Line 473-475)

Reviewer 2 Report

In this manuscript, authors present a sensorless and low-gain brushless DC motor controller using simplified dynamic force compensator for robot arm application. Authors also present experimental results for supporting their research. However, the following issues emerge:

- Why do authors claim in the Abstract, line 18, that "robot arms do not require high precision" when are applied for "safe human-machine interactions" (see line 1)? This seems to be contradictory, since authors also claim in the Introduction that "service robot emphasize mobility and safety of human robot interaction" (see lines 41 and 42) . Please clarify or modify the abstract or the introduction.

Although the paper shows an interesting topic, the motivation and contribution are poorly supported. In this sense, the reviewer considers that authors must highlight their contribution by adding recent references, since, in this version, the references are out of date.

In Figure 3 authors show "PWM signal output on current controller". Why does the PWM signal is not a pulse signal?

The dynamic model given by equation (3) does not consider perturbations. How would be the dynamic model of the robot arm if perturbations where considered? and Why do the authors not consider perturbations? This kind of systems always are subjected to perturbations generally given by parametric uncertainty. Please explain.

The description of equation (3) should specify the dimensions of each matrix and vector; since such an equation is a matrix equation and not an scalar equation.

In equation (4) variable "\tau_e" is called the electromagnetic torque. The same variable in equation (6) is now called dynamic torque. Please specify the correct name of that torque.

What is the meaning of constants "M_m" and "B_m"? Is not clear from the description given in lines 161 and 168 if those constants are scalar or constant matrices. 

Figure 6 shows a block named "Kinematics", which apparently contains only dynamic parameters. Please clarify.

According with the title of the paper, "a sensorless and low-gain brushless DC motor controller" is designed. The reviewer thinks that the SDFC corresponds to such a controller. However, authors mention in lines 253 and 254 "the host controller" and the "slave driver". In this sense, the reviewer asks that confusion for readers must be avoided; since "the host controller" should be called "the host card controller" or some.

Related to the design of the SDFC controller. Is it possible to solve the trajectory tracking task with such a controller? Because it seems that only the position task was solved.

As far as the reviewer observed, this manuscript was focused from an engineering point of view. However, it would be important that some guidelines related to the stability analysis in closed-loop be given.

Equation (17) correspond to the "Efficiency analysis" of the SDFC. However, authors never showed some numbers related to this analysis. Then, What is the purpose of including Equation (17)? The reviewer thinks that it should be eliminated.

Authors should be more careful regarding the figures, since captions associated with Figures 2, 3, and 4 were repeated. Also, in line 456 authors mention the Figure 18 and such a figure was not included in the manuscript. Hence, Figures 16 and 17 are also missed. In consequence, maybe the presented results do not match with the corresponding text and some confusion could emerge for the readers.

Author Response

Dear reviewers:

We are very grateful to the precious comments made by reviewers, which provide insightful directions for significantly improving the quality of the manuscript. According with these advices, we carefully amended relevant parts in the manuscript. The responses to the comments and questions are listed below.

Point 1:

Why do authors claim in the Abstract, line 18, that "robot arms do not require high precision" when are applied for "safe human-machine interactions" (see line 1)? This seems to be contradictory, since authors also claim in the Introduction that "service robot emphasize mobility and safety of human robot interaction" (see lines 41 and 42). Please clarify or modify the abstract or the introduction.

Response 1:

Thank you for your correction. This discussion in the Abstract was indeed not clear in meaning. We wanted to present that the service robot does not require high precision sensor for high-resolution position control. Also, the robot’s mobility and safety are more important on service applications. We have included a clearer discussion in the Abstract. (P. 1, Line 17-19)

Point 2:

Although the paper shows an interesting topic, the motivation and contribution are poorly supported. In this sense, the reviewer considers that authors must highlight their contribution by adding recent references, since, in this version, the references are out of date.

Response 2:

Thank you for your suggestion. In this study, we improved sensorless motor control technology by using dynamic compensation. The researches of sensorless motor control had been found in many studies in the past, listed in Table 1. However, these controls were never realized on a robot arm due to problems of delay and low response. (Discussed in Line 52-62, P. 2)

Therefore, we presented an open-loop dynamic compensator to solve these problems. However, the calculation of the dynamic equation involves complex multidimensional matrix operations, which requires long coding times and high costs, making it mostly infeasible on real-time robot control in reality. (Discussed in Line 72-76, P. 2)

For reducing cost, the calculation of dynamic equation was simplified on FPGA driver and the numbers of model parameter were reduced. Also, we made a real-time process of dynamic compensation using only a low-cost robot controller and FPGA drivers. (Discussed in Line 80-85, P. 3)

We have revised and updated with some recent references.

Point 3:

In Figure 3 authors show "PWM signal output on current controller". Why does the PWM signal is not a pulse signal?

Response 3:

Thank you for your correction. In Figure 3, the PWM signal must be revised to voltage signal. In order to generate an analog voltage on FPGA, we use PWM digital signal on motor voltage control. Therefore, Figure 3 presented the results of voltage signal with LPF. We have revised this discussion and corrected the title of Figure 3. (P.4, Line 124)

Point 4:

The dynamic model given by equation (3) does not consider perturbations. How would be the dynamic model of the robot arm if perturbations where considered? and Why do the authors not consider perturbations? This kind of systems always are subjected to perturbations generally given by parametric uncertainty. Please explain.

Response 4:

Thank you for your question. We are not sure about the meaning of perturbations. Does it mean external disturbance? Equation (3) just presented an ideal dynamic model of a manipulator such as found in many textbooks. Actually, disturbed torques must be considered on dynamic robot control in real environment. In our study, this unknown disturbed torque cannot be observed from the known robot model. Therefore, the closed-loop controller must perform servo control by state feedback for disturbed torque, such as Figure 9. We have rewritten the discussion of disturbance torque. (P. 10, Line 239-241)

 Point 5:

The description of equation (3) should specify the dimensions of each matrix and vector; since such an equation is a matrix equation and not an scalar equation.

Response 5:

Thank you for your suggestion. Equation (3) is n-joint robot dynamic model; q is joint vector,  is  torque vector,  is  inertia matrix,  is  vector of Coriolis and centripetal torque, and  is  vector of gravity torque. We have inserted dimensions of each matrix and vector in description of equation (3). (P. 6, Line 153-155)

Point 6:

In equation (4) variable "\tau_e" is called the electromagnetic torque. The same variable in equation (6) is now called dynamic torque. Please specify the correct name of that torque.

Response 6:

Thank you for your correction. In equation (4) and (6),  is the same meaning of electromagnetic torque. We have revised the name of torque in equation (6). (P. 6, Line 168)

Point 7:

What is the meaning of constants "M_m" and "B_m"? Is not clear from the description given in lines 161 and 168 if those constants are scalar or constant matrices.  

Response 7:

You have raised an important question. In equation (4) (Line 161), it is a BLDC motor mathematical model for single-joint. Therefore,  and  are constant gain in equation (4). Equation (6) is expanded from a single-joint to n-joint on robot arm model, so  and  are  constant matrixes of motors. We have revised the description of equation (4) and (6). (P. 6, Line 158-159 and Line 165)

Point 8:

Figure 6 shows a block named "Kinematics", which apparently contains only dynamic parameters. Please clarify.

Response 8:

In Figure 9, the green block ("Kinematics") mean calculation of forward kinematics by each joint angle position. Since this calculation must need n-joint position information from each motor, this process is real-time working on host controller (robot controller), such as Raspberry Pi 3 board. The host controller may confuse readers, so it has been revised to the robot controller. (P. 10, Line 235-238)

Point 9:

According with the title of the paper, "a sensorless and low-gain brushless DC motor controller" is designed. The reviewer thinks that the SDFC corresponds to such a controller. However, authors mention in lines 253 and 254 "the host controller" and the "slave driver". In this sense, the reviewer asks that confusion for readers must be avoided; since "the host controller" should be called "the host card controller" or some.

Response 9:

Thank you for your correction. Such as Response 8, the words of “host controller” and “slave driver” were not clear for readers in discussion. The host controller is an additional robot controller outside of FPGA drivers, such as Figure 14. We try to allow readers to easily understand the true meaning by revising to two new words: “robot controller“ and “FPGA driver”. We have revised discussion of these two words. (P. 10, Line 235-238)

Revised:

Host controller (host card controller) -> Robot controller

Slave driver -> FPGA driver

Point 10:

Related to the design of the SDFC controller. Is it possible to solve the trajectory tracking task with such a controller? Because it seems that only the position task was solved.

Response 10:

You have raised an important question. In this SDFC design, the input are velocity and acceleration commands, and position feedback, as Figure 9. The SDFC is an independent and real-time process in FPGA driver. In this study, the velocity and acceleration commands are calculated on each joint FPGA driver. Therefore, the trajectory tracking task can be used on SDFC if the tracking velocity and acceleration are real-time supported form a controller.

Point 11:

As far as the reviewer observed, this manuscript was focused from an engineering point of view. However, it would be important that some guidelines related to the stability analysis in closed-loop be given.

Response 11:

Thank you for your suggestion. We have discussed analysis of the closed-loop control performance in Section 4.3. (P. 13, Line 305-311)

In Figure 12, low-gain closed-loop servo control, such as blue line, presented position error and phase delay. It means only closed-loop controller is not enough on this application. After incorporating the SDFC, the force compensator reduced the dynamic error to less than 1° (red line). Comparing the current output results with and without the SDFC, current response was faster with compensation; this verified that the SDFC increased control response and reduced dynamic error.

Also, we have discussed compensator efficiency by closed-loop control analysis in Section 6.2.1. (P. 18, Line 443-460)

Point 12:

Equation (17) correspond to the "Efficiency analysis" of the SDFC. However, authors never showed some numbers related to this analysis. Then, what is the purpose of including Equation (17)? The reviewer thinks that it should be eliminated.

Correct number of equation: (17) to (16)

Response 12:

Thank you for your suggestion. We think the description of equation (16) is not clear for readers. The equation (16) used on results of efficiency analysis experiment, as shown in Table 9 (Current error rate E). The current error rate (E) means the error of compensation, the error rate is smaller, meaning accuracy of compensation is better. We have discussed results in Line 449-460. We have revised to get a clear discussion in equation (16). (P. 18, Line 427-429 and Line 440-442)

Point 13:

Authors should be more careful regarding the figures, since captions associated with Figures 2, 3, and 4 were repeated. Also, in line 456 authors mention the Figure 18 and such a figure was not included in the manuscript. Hence, Figures 16 and 17 are also missed. In consequence, maybe the presented results do not match with the corresponding text and some confusion could emerge for the readers.

Response 13:

We indeed have some mistakes on wrong setting in Word. Thanks for your precious suggestion. We will be more careful regarding to the figures.

Round 2

Reviewer 2 Report

The reviewer thanks the authors by considerig his comments.